# New estimates of flood exposure in developing countries using high-resolution population data

Andrew Smith[1], Paul D. Bates [1,2], Oliver Wing[1,2], Christopher Sampson[1], Niall Quinn[1] & Jeff Neal[1,2]

Current estimates of global flood exposure are made using datasets that distribute population counts homogenously across large lowland floodplain areas. When intersected with simulated water depths, this results in a significant mis-estimation. Here, we use new highly resolved population information to show that, in reality, humans make more rational decisions about flood risk than current demographic data suggest. In the new data, populations are correctly represented as risk-averse, largely avoiding obvious flood zones. The results also show that existing demographic datasets struggle to represent concentrations of exposure, with the total exposed population being spread over larger areas. In this analysis we use flood hazard data from a ~90 m resolution hydrodynamic inundation model to demonstrate the impact of different population distributions on flood exposure calculations for 18 developing countries spread across Africa, Asia and Latin America. The results suggest that many published large-scale flood exposure estimates may require significant revision.

---

[1] Fathom, The Engine Shed, Station Approach, Bristol BS1 6QH, UK. [2] School of Geographical Sciences, University of Bristol, Bristol BS8 1SS, UK. Correspondence and requests for materials should be addressed to A.S. (email: a.smith@fathom.global)

In the last 5 years, considerable efforts have been made to develop continental and global-scale flood hazard models[1–8]. When combined with data sets detailing exposure and vulnerability, such schemes can quantify flood risk, which can be used by governments, insurers, and individuals to adapt to, or mitigate, these threats. The physical processes included in the hazard models used in these risk assessments have received considerable attention from physical scientists[1,9,10], but to date the exposure and vulnerability components have not. This is concerning, as arguably we know even less about the location of people and assets, and the impact of hazards on them, than we do about the frequency and nature of the flood hazard events themselves.

Population maps are a key component of risk calculations[8,11], and are often used when downscaling coarser socio-economic data sets (e.g., Gross Domestic Product (GDP)) to the resolution of the hazard model. Calka et al.[12] have noted the importance of fine scale population density data to all classes of hazard modelling and review the wide range of data sets currently available. Population data are collected via nationally organised census studies and, for confidentiality, are typically provided at the scale of small administrative units known as enumeration areas. These enumeration areas vary in size within and between countries from ~$10^2$ m$^2$ to ~$10^4$ km$^2$, with 33 km$^2$ global average[13]. By contrast, large-scale hazard models typically run simulations over regular grids using either cartesian or spherical coordinate systems with final resolutions in the range 1–30-arc seconds (~30–900 m horizontal resolution at the equator). Gridded population density data sets at a resolution (and accuracy) commensurate with the hazard model output are therefore required when calculating flood risk. A wide range of such data sets are available including GPW (Gridded Population of the World)[14], Landscan™[15], WorldPop[16], GHSL (Global Human Settlement)[17], GUF (Global Urban Footprint)[18], and HYDE (History Database of the Global Environment)[19]. Although these products use similar input data to derive population densities, Calka et al.[12] note there is no standardised methodology for doing this. As a result, estimates of population density from the various global gridded data sets vary markedly[20].

To date, there have been a number of attempts to estimate flood exposure over large scales (see Supplementary Table 1). However, there can be large differences between the resolution at which the hydraulic computations are performed (1 arc second to 0.5 degrees), the resolution of the hydraulic model output after downscaling (1–30-arc seconds), the resolution of the gridded population data (1 arc second to 5 arc minutes) and the resolution at which the final exposure calculations are performed (1 arc second to 5 arc minutes). Almost the full range of population data sets noted above have been employed, however studies to date do not report flood exposure estimates consistently, with this variation remaining large even when broadly consistent metrics are being used[3,7,21]. Although the choice of population data only accounts for part of the differences between these complex modelling systems, it is clear that the effect of this choice, and its subsequent treatment, on the output of exposure calculations needs further investigation.

Global and continental scale flood inundation models now operate at resolutions of 1–3-arc seconds (~30–90 m) and show good skill in hazard prediction at these scales, with Critical Success Index values up to 0.7 and Hit Rates up to 90%[5,8]. Local flood models operate down to scales of a few metres[22–24], and show even higher levels of skill with pixel-scale Critical Success Index values up to 0.9. However, the population data sets used to date remain relatively coarse; even the ~3-arc second (90 m) resolution WorldPop data only disaggregates census data to settlements identifiable in medium resolution (30–90 m) satellite imagery[25], which effectively means only settlements a few hundred metres across can be conclusively discriminated. Thus, its true intrinsic resolution is likely to be somewhat lower than the stated (i.e., nominal) resolution of ~90 m. Owing to the very local nature of flooding and the propensity for humans to avoid harm, any degradation of resolution in the hazard or population data will likely lead to an increase in exposure estimates. There is, therefore, a clear need for accurate population estimates at resolutions commensurate with the latest generation of flood hazard models.

Very high-resolution (sub-metre scale) satellite imagery provides a potential solution to this problem. These data can be used to identify habitation centres down to the building level, enabling the disaggregation of census data with greater fidelity. Moreover, recent developments in artificial intelligence permit a step change in the resolution and accuracy of country-scale population density mapping.

In this paper, we use new high-resolution (1 arc second, ~30 m) population density to map flood exposure for 18 countries[26]. Tiecke et al.[26] describe this new population data set, called the High Resolution Settlement Layer (HRSL) reporting significant skill in representing individual buildings and also marked improvements in representing rural populations. Moreover, a separate validation of the HRSL was conducted and is included in the λ (HRSL Validation in Supplementary Information). This validation procedure concluded that the HRSL population data has considerable skill in identifying building footprints. These data are derived from imagery capable of resolving individual buildings the true (intrinsic) resolution will be close to the stated (nominal) value, and commensurate with the latest generation of large-scale flood hazard models. We intersect the high-resolution population data with 3-arc second (~90 m) flood depth data produced by a global-scale true hydrodynamic model[27,28], to produce estimates of the population exposed to flooding. We compare the estimates of the populations exposed to flooding derived in this way to estimates using two further global population data sets: WorldPop (3-arc second, ~90 m)[29] and LandScan™[30] (30-arc second, ~900 m). We also use the Global Human Settlement Layer[31], produced by the European Joint Research Centre, to explore how estimates of flood risk vary across rural, semi-urban and urban areas. The research aims to explore the implications of emerging demographic data sets on our current understanding of flood exposure.

## Results
**Exposure calculations.** Estimates of flood exposure, derived using a comprehensive ~90 m resolution hydraulic modelling framework and differing population data sets, reveal a universal bias in the calculations made using WorldPop and LandScan™ data, in comparison with the estimated returned when using the HRSL data. Across each of the 18 countries included here, estimates of the population exposed to a 1 in 100 year flood (the area of land with a 1% chance of being inundated in any given year) are smaller when HRSL population data are used instead of WorldPop or LandScan™ data. Table 1 shows the total population exposed across each territory and for each population data set. The reduction in exposed population when the HRSL data are used to define population distribution can be as much as ~60%; in Uganda, exposure totals reduce from ~4 M to 1.66 M when WorldPop and LandScan™ data are replaced by HRSL. Overall, the total population exposed to the 100 year flood in the 18 countries was calculated to be 134, 122, and 101 million when using the LandScan™, WorldPop, and HRSL data, respectively.

Figure 1 shows the cumulative distribution of exposed population across all of the modelled inundated area. The figure

| Table 1 Population located in the 1 in 100 year floodplain (millions) | | | | | |
|---|---|---|---|---|---|
| Country | WorldPop | LandScan™ | HRSL | WorldPop change % | LandScan™ change % |
| Burkina Faso | 2.40 | 2.72 | 1.74 | −27 | −36 |
| Cambodia | 6.26 | 6.86 | 4.69 | −25 | −32 |
| Ghana | 3.17 | 3.83 | 2.57 | −19 | −33 |
| Haiti | 3.14 | 3.22 | 3.09 | −1 | −4 |
| Madagascar | 4.41 | 4.66 | 3.50 | −21 | −25 |
| Malawi | 2.54 | 2.50 | 1.61 | −37 | −36 |
| Mexico | 30.36 | 30.08 | 24.37 | −20 | −19 |
| Mozambique | 5.69 | 6.38 | 3.76 | −34 | −41 |
| Philippines | 43.86 | 50.12 | 42.65 | −3 | −15 |
| Puerto Rico | 0.81 | 0.80 | 0.68 | −16 | −16 |
| Rwanda | 0.95 | 1.03 | 0.59 | −37 | −42 |
| South Africa | 3.39 | 4.86 | 2.02 | −41 | −59 |
| Sri Lanka | 3.62 | 4.52 | 2.84 | −22 | −37 |
| Tanzania | 7.72 | 7.86 | 5.29 | −32 | −33 |
| Uganda | 4.18 | 4.56 | 1.66 | −60 | −64 |

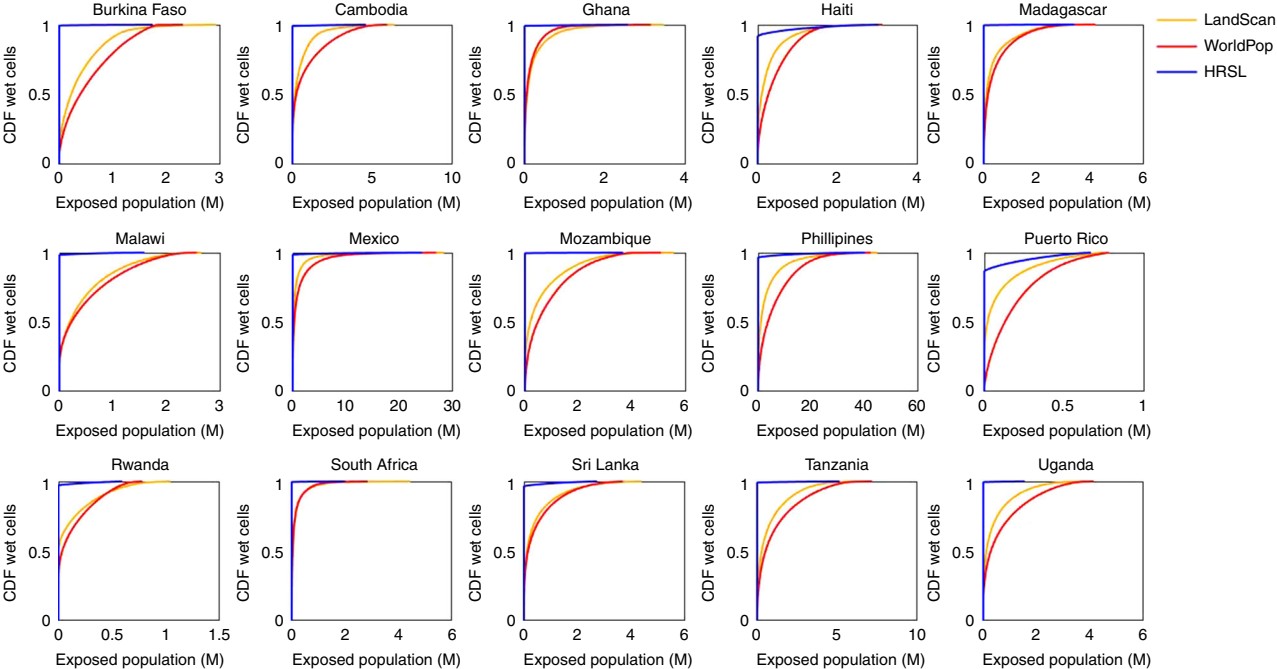

**Fig. 1** Distribution of exposed population. Cumulative distribution of the population living in the 1 in 100 year floodplain distributed across all the modelled floodplain cells. Red indicates WorldPop exposure, Yellow the LandScan™ exposure, and Blue the exposure calculated using the HRSL data

reveals that the distribution of exposure is markedly different between the HRSL data and the other two population data sets. Across all regions, the majority of the modelled inundation is in areas where WorldPop and LandScan™ data indicate populations are situated. Conversely, when the HRSL data are used, only a very small proportion of modelled wet cells are in areas with a non-zero population count, meaning exposure is spread over a far smaller proportion of the hazard area. For example, in Malawi, ~80% of modelled wet cells overlay inhabited areas according to the WorldPop and LandScan™ population data, compared with only ~2% when the HRSL population data are used. This phenomenon involving the total population exposed being spread across a far larger number of modelled wet cells is found across all countries studied, indicating that even in countries where exposure totals are similar between each population data set, the concentration of exposure is markedly different. This is evident when looking at the exposure totals calculated for Haiti. Here, estimates of total population exposed to flooding range between 3.14 M and 3.09 M for the WorldPop and HRSL derived estimates respectively, constituting a small change of −1%. However, in the case of the WorldPop data this exposure is spread over an area of ~40,000 km² , compared with an area of ~3700 km² when using HRSL data. Figure 2 further highlights these differences in exposure concentration, displaying population data and the resulting exposed population for the area around Lilongwe city, in Malawi. First, there are significant differences in the way populations are mapped, with the WorldPop and LandScan™ data returning a non-zero population density across almost the entire region. When these populations are intersected with the 1 in 100 year hazard data, the result is that almost all the modelled wet cells in the region generate exposure (Fig. 2e, f). Conversely, the HRSL demographic data distribute populations across a far smaller area, resulting in much higher population densities. When these data are intersected with the 1 in 100 year hazard data, the majority of the modelled hazard area does not generate exposure (Fig. 2d). However, far higher

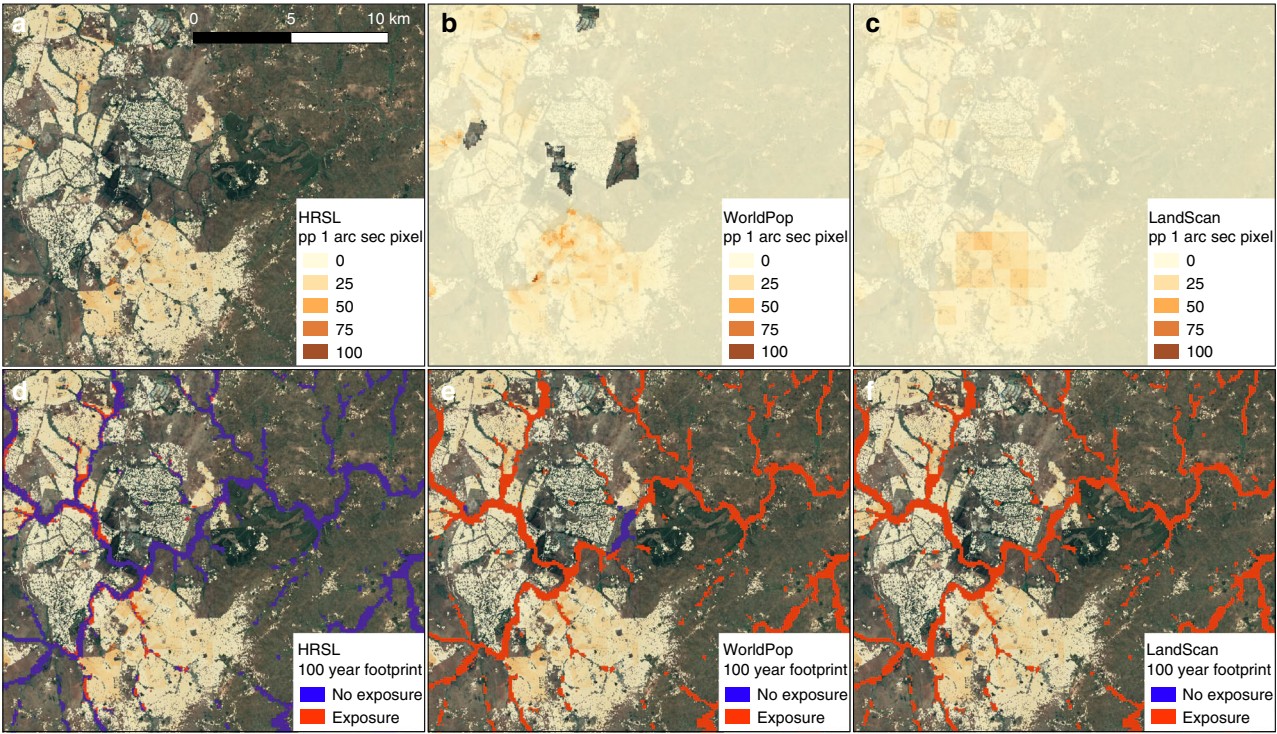

**Fig. 2** Example of Exposure Mapping in Malawi. Example of population data and generated exposure maps for Lilongwe city, in Malawi. **a**, **b**, **c** display the HRSL, WorldPop, and LandScan™ population data sets, respectively. **d**, **e**, **f** show the population located in the 100 year floodplain for each demographic data set, respectively

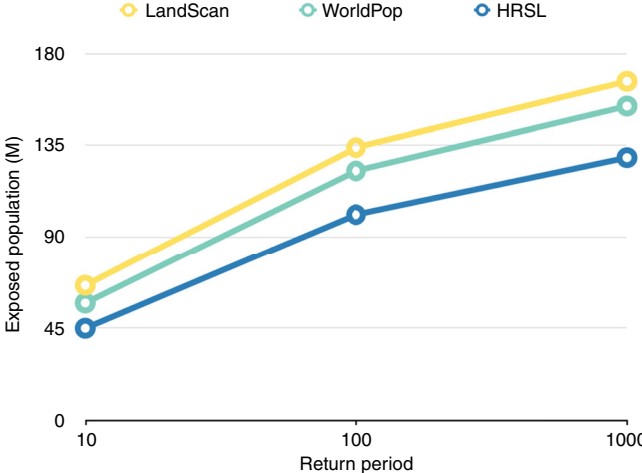

**Fig. 3** Total exposed population. Total population exposed (millions) across all 18 countries, for the 1 in 10, 1 in 100, and 1 in 1000 year flood events

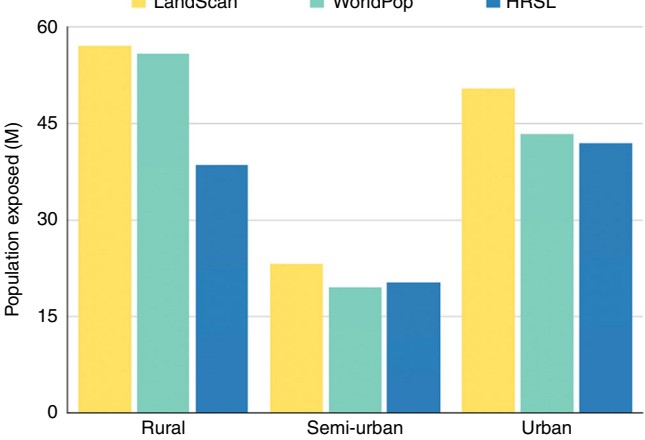

**Fig. 4** Exposure across different land use types. Total population exposed to flooding across rural, semi-urban, and urban areas (millions)

concentrations of exposure emerge which, when summed, produce total values broadly comparable to those returned by the use of WorldPop data.

**Exposure across land use types**. Further analysis, estimating flood exposure for the 1 in 10 and 1 in 1000 year flood events, show that the results found for the 1 in 100 year flood event are consistent across multiple return periods, with the HRSL consistently returning the lowest total exposure (Fig. 3). Indeed, the results show that as hazard intensity increases, the discrepancy between the HRSL results and the other population data increase.

Analysis of the population exposed to flooding across urban, semi-urban and rural area (see Methods) revealed that both the WorldPop and LandScan™ data sets estimate rural populations to be most exposed; exposed rural populations make up 47% and 44% of the total population exposed for WorldPop and LandScan™ data, respectively (Fig. 4). Conversely, the HRSL data estimate that urban populations drive the majority of the exposure, constituting 42% of the total population exposed. Figure 4 reveals that in terms of total population exposed, estimates in semi-urban and urban areas are broadly similar across all population data sets, with the majority of the positive bias in WorldPop and LandScan™ estimates being derived in rural areas. The results also show that across urban, semi-urban, and rural areas, the coarser resolution LandScan™ data returns the

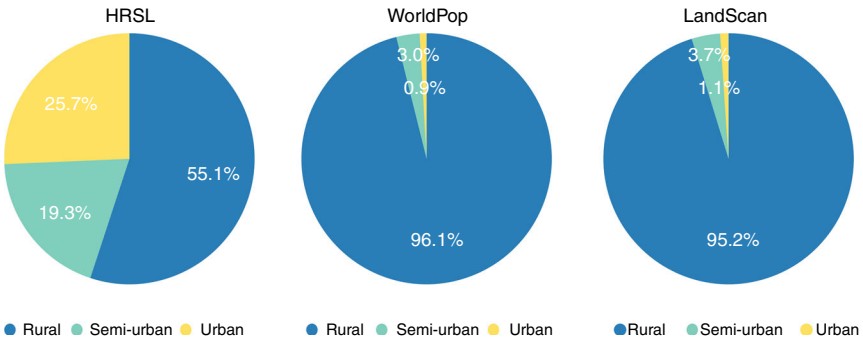

**Fig. 5** Proportion of areas returning exposure. Proportion of area returning an exposure value, across rural, semi-urban, and urban areas

**Table 2 Total population living in the 1 in 100 year floodplain (millions) summed across all 18 countries, for varying resolutions of both hazard and population data**

| | | Population data | | | | | |
|---|---|---|---|---|---|---|---|
| | | 30 m | 90 m | | 900 m | | |
| | | HRSL | HRSL | WP | HRSL | WP | LS |
| **Hazard** | **90 m** | 101 | 102 | 122 | 124 | 130 | 134 |
| | **900 m** | 196 | 196 | 205 | 197 | 205 | 203 |

highest exposure totals. Indeed, the results indicate a negative correlation between the resolution of demographic data and the total population exposed to flooding. In addition to contrasting exposure values, Fig. 5 reveals significant differences in the areas deriving flood exposure, with > 95% of the total exposed area (area where both hazard and population data are non-zero) being rural in nature for the WorldPop and LandScan™ data sets, whereas only ~1% of the total exposed area was defined as being urban. In the case of the HRSL data, more than a quarter of the total exposed area (25.7%) was defined as being urban and only 55.1% rural. These results further emphasise the overestimation of flood exposure in rural areas that arises when existing demographic data sets are used.

**Exposure calculations at different resolutions**. Table 2 presents the results of a sensitivity analysis, varying the resolution of both hazard and population data. When all population data are aggregated to ~900 m horizontal resolution, the total population exposed is > 120 million. Coarsening of the hazard data reveals a larger increase in total exposure; when the hazard data is aggregated to ~900 m resolution, the total population exposed is ~200 million regardless of the resolution of the population data. Finally, for each of the aggregated resolutions, the HRSL returned the lowest estimate of total population exposed.

**Discussion**
The results presented here indicate that analyses of the number of people exposed to flooding undertaken with existing population data sets may be significant overestimates. This finding was found to be consistent across multiple return periods, suggesting that exposure estimates conducted using the HRSL data are not more sensitive to changes in the magnitude of hazard intensity when compared with the results from the WorldPop and LandScan™ data. Consequently, the findings presented here are expected to be robust to uncertainties in the underlying hazard data. The results also highlight the importance of using highly resolved flood hazard data to conduct estimates of flood exposure at large scales;

only a combination of both high-resolution population data and high-resolution hazard data results in exposure reductions. Moreover, the results suggest that exposure estimates are particularly sensitive to the resolution of the underlying hazard data; with a coarsening of the hazard data resulting in a near doubling of the total population exposed (Table 2). Even when the HRSL data are aggregated to the resolutions of WorldPop and Land-Scan™ data, we see similar trends as those detected when calculating exposure at their native resolutions. This suggests that the conclusions drawn are not solely artefacts of the differing resolutions between the population data, but that the HRSL data are intrinsically more accurate. Aside from the differences in total exposed population, the results also show how concentrations of exposure vary markedly between the different population data sets. Although the total exposed population calculated using the new HRSL is lower across all countries, this exposed population is also confined to a much smaller area. Moreover, the results suggest that existing population data sets significantly over-estimate rural flood exposure, with only a very small proportion of the total exposed area being urban in nature. Critically, it is exposure hotspots in new urban centres that appear to be driving increased flood losses in developing regions; Di Baldassarre et al.[32] report that rapid and intensive urbanisation in flood-prone areas is driving dramatic increases in flood exposure across Africa. This largely unplanned encroachment onto floodplain areas creates concentrations of exposure, resulting in an amplification of losses when flood events occur. The results shown here suggest that pre-existing demographic data sets struggle to represent the significance of these exposure hotspots. This mis-estimation of exposure concentration would have significant implications for decision makers looking to use these data for adaption and mitigation. With increasing flood losses being largely attributed to increased urbanisation, it seems crucial that these areas are correctly represented.

Differences in the distribution of exposure reflect differences in the methods used to disperse populations, with census population data being spread across a far wider area in the WorldPop and LandScan™ data. Nevertheless, the results clearly demonstrate the significance of resolving populations at resolutions that are commensurate with emerging high-resolution global flood models[7,9,10]. The latest large-scale flood hazard models can resolve flood hazard at resolutions of ~30–90 m, which can be considered sufficient to allow the representation of the true spatial complexity of flood hazard[8]. However, to move from estimates of flood hazard to estimates of flood risk, data that represent the spatial heterogeneity of people and assets with commensurate resolution and accuracy are also clearly required. Otherwise, as demonstrated here, inaccurate representations of exposure may persist, rendering the output of increasingly complex flood hazard models ineffective as decision-making tools. Overall our results

show that estimates of flood exposure undertaken using existing population data may significantly mis-represent these quantities. Calculations with new high-resolution population data resulted in exposure reductions across all countries analysed. These results suggest that, in terms of flood exposure, human populations are more risk-averse than current demographic data suggest, with populations largely avoiding the most hazardous areas. However, the results also demonstrate that concentrations of exposure vary markedly. The HRSL returns larger concentrations of exposure, suggesting that although there is a reduction in the total number of people exposed, these populations are confined to a smaller area. Moreover, exposure calculations undertaken with existing demographic data may substantially overestimate flood risk in rural areas and produce underestimates in urban centres. The results have significant implications for any end-users looking to use emerging large-scale flood risk data sets to inform decision making and suggest that existing estimates may require significant revision. Moving forward, alongside the development of increasingly complex hazard models, the development of accurate data sets defining the location of people and assets will also be required if robust estimates of risk are to be generated.

## Methods

**Flood hazard model.** A global flood hazard model framework was used to define flood hazard across 18 developing countries; Burkina Faso, Cambodia, Ghana, Haiti, Madagascar, Malawi, Mexico, Mozambique, Philippines, Puerto Rico, Rwanda, South Africa, Sri Lanka, Tanzania, and Uganda. This framework covers both fluvial (riverine) and pluvial (flash-flood) perils, providing estimates of flood hazard at 3-arc second (~90 m) resolution. Fluvial flooding is simulated in all river basins with upstream catchment areas larger than 50 km$^2$, whereas pluvial hazard is captured across all catchment sizes via the simulation of intense rainfall directly onto the modelled topography. River channel location and bathymetry are derived from the HydroSHEDS global hydrography data set[33]. A sub-grid hydraulic model[28] is employed enabling all channels, including those smaller than the ~90 m resolution of the model, to be explicitly represented using a computationally efficient local inertial formulation of the shallow water equations[27]. Coupling a remotely sensed hydrography data set with a sub-grid hydraulic model enables the comprehensive representation of flooding from river channels across all areas, including data-poor regions. Model input boundary conditions are derived from a regionalised flood frequency analysis conducted at the global scale[34]. In principle, this method links river and rainfall gauges to upstream catchment characteristics and local climatology respectively, with gauged regions linked to un-gauged areas using these descriptors. A full description of the hazard modelling framework used here, along with a model validation, is presented by Sampson et al.[5] The study reports that, in a validation exercise comparing model output against high-resolution government data in the UK and Canada, the modelling framework captured between two thirds and three quarters of the area determined to be hazardous. A further validation study, conducted by Wing et al.[8], reported that a large-scale modelling framework similar to that used in this study was capable of matching high-quality flood hazard data in the United States to within the likely error of local scale models.

**Population data.** In this study, exposure is defined by the intersection of hazard (flood model output) and population data, the way in which hazard and population data interact (vulnerability), to produce estimates of risk, has not been considered. To estimate the number of people exposed to flooding, flood hazard data from the hydraulic model framework described above was intersected with three different population density maps, these were: the HRSL[35], WorldPop[29], and LandScan™[30]. The HRSL is a new population data set produced jointly by Facebook, Columbia University and the World Bank. Unlike the data sets produced by WorldPop and LandScan™, which use multi-variate models to disaggregate census population data, HRSL utilises cutting edge convolutional neural networks to identify individual buildings from high-resolution satellite imagery. Population census data are then distributed among these buildings to produce population density maps. The final data set is a 1-arc second (30 m) resolution population density map for the year 2015. Tiecke et al.[26] outline a number of validation exercises for the HRSL data, including the testing of building identification using the Malawi Third Integrated Household Survey (IHS3). This survey recorded the location of > 11,000 households nationwide and is thus independent of remote-sensing methods. When used as a validation data set, the results revealed that 98.3% of IHS3 household locations coincided with HRSL populated cells/pixels. A separate analysis of 3 different population data sets was undertaken for a single region near Blantyre, Malawi, where buildings were manually identified and labelled. A comparison of HRSL, GUF, and GHSL revealed that in urban areas 99%, 82%, and 83 % of buildings were identified correctly. However, in rural areas the percentages of buildings identified were 82%, 6%, and 4% for HRSL, GUF, and GHSL data sets, respectively.

The results indicate that HRSL has a far superior performance in rural areas, where existing data sets perform poorly. A separate validation of each of the population data sets was also undertaken as a part of this study (see HRSL Validation in Supplementary Information). This procedure compared each population data set against building footprints taken from the Open Street Map (OSM) project (Supplementary Table 2). The comparison concluded that the HRSL data have considerable skill in replicating OSM data (Supplementary Fig. 1), whereas the WorldPop and LandScan™ data had little to no skill in replicating building footprints (Supplementary Tables 2–5). Population density maps provided by WorldPop use remotely sensed data in a dasymetric modelling approach to estimate population densities at 3-arc second (~90 m) resolution[16]. This method uses a range of remotely sensed data sets, including night-time light data and water surface masks, to produce a prediction layer defining the likely population distribution. The data sets used to produce the prediction layer vary between different regions, with some input data sets being produced at a resolution coarser than the stated 3-arc second resolution. This population prediction layer is used to distribute census population data, with the 2015 population density maps being used in this study. The coarsest resolution (30-arc second, ~900 m) population density data used here was provided by LandScan™ [30]. These data were also produced using a multi-variable dasymetric modelling approach to disaggregate census data within administrative boundaries. Similar to the WorldPop methodology, the method uses a range of input data sets to distribute census information, with input data and methods varying between different regions.

To estimate the total number of people living in floodplain regions, the hazard layers were intersected with HRSL, WorldPop, and LandScan™ data. To enable this, each population data set was disaggregated to the ~30 m resolution of the HRSL data set. This disaggregation was conducted by taking the population totals at a coarser resolution and distributing them among the higher resolution cells. To ensure consistency in the total population values between the different demographic data sets, both the WorldPop and LandScan™ data were scaled and the country level to ensure that population totals matched the HRSL population totals. Intersection with the hazard data involved summing pixel values from the population map for all wet cells in the hazard map (i.e., all cells with a water depth greater than zero). To enable a sensitivity analysis across different resolutions both the HRSL and WorldPop data had to be aggregated to coarser resolutions. This aggregation was undertaken by taking the sum of the population present in the grid cells at a higher resolution. Aggregation of the hazard data was undertaken by taking an average of all higher resolution cells and applying a 10 cm depth threshold to all resulting depth calculations, whereby all cells with a depth < 10 cm are set to zero. This depth threshold is the same threshold applied directly to the ~90 m flood hazard model output. To enable an analysis across different land use types, the Global Human Settlement Layer[31] produced by the European Joint Research Centre was used to define urban, semi-urban, and rural areas.

## Data availability

Each of the population data sets used here are available via the relevant references provided. The HRSL can be found at https://www.ciesin.columbia.edu/data/hrsl/. The flood hazard data used are available for academic use, for more information contact info@fathom.global. The code for the hydraulic model is owned by Fathom Ltd and is not available to be shared.

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

## Acknowledgements

Paul Bates was supported by a Research Fellowship from the Leverhulme Trust and a Royal Society Wolfson Research Merit Award. Jeff Neal was supported by UK Natural Environment Research Council grant NE/S006079/1.

## Author contributions

A. Smith lead the study along with P. Bates. Data processing and analysis was assisted by O. Wing and N. Quinn. Processing of hydraulic model data was undertaken by C. Sampson. Validation of the exposure data was led by J. Neal. All authors assisted in writing the manuscript.

## Additional information

**Competing interests:** The authors declare no competing interests.

