## [Peer Review File · Nature Communications]

Reviewers' comments:

Reviewer #1 (Remarks to the Author):

Applying Convolutional Neural Network techniques and 0.5 m resolution satellite imagery, the authors of the article present a method of using new high resolution (1 arc second, ~30 m) population density data to map flood exposure in 18 African, Asian and Latin American countries. They compare obtained this way estimates of flood-exposed populations with two further global population datasets: WorldPop and LandScan. The authors also use the Global Human Settlement Layer, produced by the European Joint Research Centre, to find out how estimates of flood exposure vary across rural, semi-urban and urban areas.

The main claim of the article is that the results of the analysis clearly suggest that many large-scale flood risk estimates published to date may require significant revision. There is a clear need for accurate population estimates at resolutions commensurate with the latest generation of flood hazard models. Nowadays, this topic is very important, but still needs to be developed, which is why this work can be considered for publication. The paper is well written and the authors clearly state what motivated them to do the research. The article fits the scope of Nature Communications and will be of interest to its readers. However, there are some minor deficiencies which should be corrected before the final publication:

1. It seems that a summary sentence expressing the purpose of the paper should be added to the Introduction.
3. The use of the same colours, blue and green, in Figure 3 and Figure 4 might be misleading for the reader. I suggest changing the colours in Figure 4.
4. There is a discrepancy between page 8, line 117-118: "Uganda, exposure reduces from ~4M to 1.55M when WorldPop and LanScan data are replaced by HRSL" and the data for Uganda in Table 2. This needs correcting.
5. There is a discrepancy between page 9, line 134-135: „Here, estimates of total exposed population range between 3.12M and 3.01M for the WorldPop- and HRSL- derived exposure estimates respectively, constituting a small change of -4%" and the data for Haiti in Table 2: 3.15M and 3.11M. This needs correcting.
6. Scale bars are missing in Figure 2.

To sum up, the article is of high scientific value and its publication in Natural Communications would be advisable.

Reviewer #2 (Remarks to the Author):

Dear Editor and Authors,

I read with much interest the manuscript by Smith and colleagues: "New estimates of flood exposure in developing countries using high-resolution population data". The authors provide a very important and novel contribution towards advancing our understanding of major uncertainties in current estimates of the number of people that are exposed to (and/or ultimately at risk of) extreme flooding. They do so by investigating limitations in accuracy and resolution in existing population distribution maps. Their efforts need to be applauded, and the results are quite stunning regarding the amount of uncertainties they found, which will likely have major ramifications for existing and future studies relying on these kinds of data. This issue has been speculated about in the past, but, to my knowledge, this study provides the most convincing quantification of the problem at a global scale. The findings would represent a major improvement in our attempts to produce reliable information for flood mitigation and management strategies.

That being said, I am afraid I cannot support publication of the manuscript in its current form, in particular in such a high-level journal as Nature Communications, for one major reason: The entire study hinges on the assumption that the new High Resolution Settlement Layer (HRSL) is superior in quality to any other global population dataset. However, there is very little (if any) evidence provided in the manuscript to support this assumption. The text only states that "The final [HRSL] density maps were validated against ground-truth household surveys taken from the World Bank Living Standards Measurement Study (LSMS)" (lines 265-267), but no specifics or quantitative results from this comparison are mentioned. I tried looking up this dataset and could not find any relevant scientific publication regarding its validity. All I could find was a short project description on a Facebook blogpost from 2016 (<https://code.facebook.com/posts/596471193873876>), stating that: "Our collaborators at the World Bank validated the Malawi dataset and found that on average we miss only about 6% of the houses that are part of their 2011 household survey data." This was followed by a broken link and I could not find any additional information. I am afraid that the lack of presented validation of HRSL (either by using citations or by own efforts by the authors) renders the study not publishable in its current form. Please do not misunderstand: I am NOT assuming that HRSL is wrong; I actually believe it is very likely the better dataset. But this needs to be shown with proper scientific evidence rather than just be assumed.

Here is as an example of why I think a solid verification of HRSL is indispensable: As Smith et al. show, a main feature of HRSL seems to be that it registers far lower population densities in rural areas, i.e. mostly zero, while other datasets show mostly small but non-zero values in rural areas. It might be possible that this is, at least in part, caused by the detection methods: HRSL is based on remote sensing imagery from which individual buildings are detected, and this may lead to higher uncertainties in very low density areas. With zero values in rural areas in HRSL, and assuming that the total population numbers for districts or countries are kept the same as in other datasets, the total population would shift towards urban areas by default. This shift of rural to urban population (true or not) would explain some of the results presented in the study by Smith et al. Hence testing whether the zero rural population numbers are indeed correct in HRSL is a fundamental issue for the validity of the presented study.

There are a two other larger issues I would like to mention, yet they seem easier to be addressed:

a) Lines 144-145: "HRSL demographic data distributes populations across a far smaller area, resulting in much higher population densities". And lines 221-224: "The HRSL returns larger concentrations of risk, suggesting that although there is a reduction in the total number of people exposed, these populations are confined to a smaller area, constituting higher concentrations of exposure." This is a very interesting characteristic of the HRSL data and indicates that likely HRSL will show a strong sensitivity when intersected with flood risk pixels; i.e. a single pixel of additional

inundation extent can trigger an enormous addition of population exposed (which is not the case if population data is more smoothly distributed). This raises the question whether the application of HRSL data can lead to increased total uncertainties when intersected with the inherent spatial uncertainty of inundation data. It would be very interesting to conduct some kind of sensitivity analysis by applying uncertainty boundaries in the inundation data and see what the effect is in the total exposure results. This is particularly relevant as the manuscript states that the employed flood modelling framework “captured between two thirds and three quarters of the area determined to be at risk” (lines 252-254). What if the missing third or quarter of flood areas has high population singularities in them? Would this increase the numbers of total exposed population to be similar to those of traditional population datasets?

b) In the methods section, it is not explained how the intersection between the 30m pixels of the flood model and the coarser 90m and 900m pixels of the WorldPop and LandScan population datasets was conducted. Line 284 only indicates that population values of wet pixels were summed. Were the coarser pixels first disaggregated to the finer 30m resolution? If no, this would be a major problem. If yes, how was the disaggregation conducted? In particular: was it assumed that population density is homogeneous within each coarser pixel? And if yes, what implications would such an assumption have? This could be tested by re-calculating the results, yet assuming that population numbers within the coarser resolution pixels are NOT equally distributed within the pixel, but people are located exclusively in the non-flood parts of the pixel. This could easily be achieved by simply removing all original WorldPop and LandScan pixels from the analysis that are only partially flooded. If this would lead to results similar to those delivered by HRSL, then the problem is not the accuracy of WorldPop and LandScan data, but only their coarser resolution (whereas currently the authors seem to indicated that the problem lies in both accuracy and resolution).

And here are some additional minor/detailed comments:

- The text is generally well written and clear, but could be a little more concise in parts. For any revision or new submission elsewhere, I encourage the authors to carefully check the manuscript and remove any redundant or repetitive statements, sentences, or parts of sentences.
- The definitions of exposure, vulnerability, risk, and hazard are not entirely clear in the manuscript. It seems that sometimes “exposure data” is used synonymous with “population data”. A bit more explanation would be helpful.
- Line 24: you say that pixels at a resolution of 1-30 arc seconds have an area of ~30-900 square meters at the equator. You need to square the numbers if you provide results in square meters, i.e. 1-30 arc seconds equals 900-810,000 square meters.
- Line 84: Replace “of volume” with “volume”
- Line 120: The HRSL data is suddenly called “Facebook data”. Be consistent.
- Line 122 (and possibly elsewhere): the use of the word “significantly” implies some statistical metric has been used (i.e. statistically significant), but I guess this is not the case here; use other terminology to avoid confusion, or provide the statistical metrics.
- In Table 2, the HRSL column is presented twice. This could be avoided if HRSL becomes the first column, and then WordPop and LandScan numbers are provided, each with absolute numbers followed by percent difference.
- Figure 2: Legend of panels A-C is confusing. It seems on panel A there are a lot of NoData pixels. How is NoData different from the lowest color (yellow) which seems to indicate zero population? I guess the legend statement “0” does not really mean zero but some range that is larger than zero? And the other colors represent ranges as well?
- Line 157 (and elsewhere): I am not sure whether the distinction of urban vs. rural areas should be simply labeled as “land use types” (I typically expect many more classes in land use classifications, e.g. agricultural techniques, recreational use, etc.). Is there a better expression?
- Lines 168-174 (and Figure 4): I guess what you describe here as “exposed” areas are areas where inundation AND non-zero people intersect in a pixel, right? This is not entirely clear as the land area would still be exposed even if there are no people living there.

- Lines 190-192: You say that "Critically, it is exposure hotspots in new urban centres that appear to be driving increased flood losses in developing regions." I cannot see any evidence in your study that supports this statement. Is this a literature citation? If not, you need to explain where this statement comes from.
- Line 217: Replace "spatially" with "spatial"
- Line 215: Remove "t"
- Line 265: Replace "produced" with "produce"

Reviewer #3 (Remarks to the Author):

Review of "New estimates of flood exposure in developing countries using high-resolution population data" by Smith et al.

I really enjoyed reading this paper and find that it has valuable contributions to our understanding of flood exposure estimates. It provides detailed information on the total exposure of people in of 15 developing countries at an unprecedented spatial resolution. Clearly the HRSL population data provides additional information that is needed to compute the flood exposure. Overall, this is a nice manuscript, but I have one major point of concern and that is towards the degree of novelty of this work.

In the beginning of the manuscript the authors give the impression that they developed a 90m population dataset and then used their flood model to give flood exposure estimates. After careful reading I find that the map is created by Facebook, Columbia and the World Bank. This clearly takes away most of the novelty of this work, since now the authors use their existing high-resolution flood model and overlay it with an new population map. Although I can see that this provides a novel dataset that has not been created before (as correctly addressed by the authors in Table 1) it is not extreme (technically) difficult and/or novel. Previous studies did the same experiment but then at coarser spatial resolution. The HRSL population data is clearly novel and so is the exposure map, but it can be seen as incremental science compared to earlier flood exposure estimates by the same group (<http://www.fathom.global/flood-maps>, scroll down toward the bottom). They already have their 90m flood model (with flood defenses as I can find on the website, not in the paper, maybe add this information or I missed it) and then it is a simple overlay operation. Maybe I'm completely wrong, in that case I think the authors would find it easy to convince me and potential reader.

Although I state above that I have some problem with the degree of scientific novelty, I can clearly see an add value for the world community. If the authors agree with me that this is the major selling point of the manuscript, I feel that this analysis can easily be done at the global scale. The authors already have the global flood maps at 90m, they have the HRSL population data at global scale, so I see no good reason why they shouldn't do that. This would also give nice figures (fancy Nature world map figures), and can also inform the audience on where we have currently under/overestimated

the flood exposure. The authors indicate that the largest difference can be found in the rural areas, so I would expect substantial reductions in flood exposure in, for example, China.

To summarize, it is a sound manuscript, clearly written and a good story. I can see the added value of this manuscript, but I would just hope that it would provide either some more novel computation or analysis to make it a Nature Communications paper. I have provided some suggestions below:

- Do a global analysis
- Perform the same analysis (if possible with the flood model input), for different return periods
- Perform the same analysis with and without flood defenses
- Perform the same analysis but then for two different time periods. One historical vs current population, showing an increase or decrease in exposure

Minor comments:

- It should be stated more clearly that the HRSL dataset is an existing dataset, the manuscript now give the impression that it is created in this manuscript.
- Line 120, there is a Facebook reference, which should be HRSL
- Line 166-167, I'm not entirely sure if we should call this a negative correlation, since it only has three points on which this statement is based (the three different datasets). Maybe rephrase to negative relationship
- Did the author validate the population or flood model? Is there a way to verify if the population map is accurate and or if the flood maps are accurate? I can see that it can be difficult to validate the flood maps, but are there some partial validations available?
- Figure 1, could be improved by using a log scale or something, now it is difficult to see the blue line and the divergence between the lines for the high exposure values
- Maybe the author can make a world map figure with a pie chart per country? Then the reader has a visual on where the biggest changes occur

Response to Reviewers Comments

Red = Author comments

Black = Reviewers comments

Firstly, we would like to thank all the reviewers for their comments. Their suggestions and edits have undoubtedly resulted in a stronger piece of research.

Reviewer 1

Applying Convolutional Neural Network techniques and 0.5 m resolution satellite imagery, the authors of the article present a method of using new high resolution (1 arc second, ~30 m) population density data to map flood exposure in 18 African, Asian and Latin American countries. They compare obtained this way estimates of flood-exposed populations with two further global population datasets: WorldPop and LandScan. The authors also use the Global Human Settlement Layer, produced by the European Joint Research Centre, to find out how estimates of flood exposure vary across rural, semi-urban and urban areas.

The main claim of the article is that the results of the analysis clearly suggest that many large-scale flood risk estimates published to date may require significant revision. There is a clear need for accurate population estimates at resolutions commensurate with the latest generation of flood hazard models. Nowadays, this topic is very important, but still needs to be developed, which is why this work can be considered for publication. The paper is well written and the authors clearly state what motivated them to do the research. The article fits the scope of Nature Communications and will be of interest to its readers. However, there are some minor deficiencies which should be corrected before the final publication:

1. It seems that a summary sentence expressing the purpose of the paper should be added to the Introduction.

Added.

3. The use of the same colours, blue and green, in Figure 3 and Figure 4 might be misleading for the reader. I suggest changing the colours in Figure 4.

Corrected.

4. There is a discrepancy between page 8, line 117-118: "Uganda, exposure reduces from ~4M to 1.55M when WorldPop and LanScan data are replaced by HRSL" and the data for Uganda in Table 2. This needs correcting.

Corrected.

5. There is a discrepancy between page 9, line 134-135: „Here, estimates of total exposed population range between 3.12M and 3.01M for the WorldPop- and HRSL- derived exposure estimates respectively, constituting a small change of -4%" and the data for Haiti in Table 2: 3.15M and 3.11M. This needs correcting.

Corrected.

6. Scale bars are missing in Figure 2.

Corrected.

To sum up, the article is of high scientific value and its publication in Natural Communications would be advisable.

Reviewer 2

I read with much interest the manuscript by Smith and colleagues: “New estimates of flood exposure in developing countries using high-resolution population data”. The authors provide a very important and novel contribution towards advancing our understanding of major uncertainties in current estimates of the number of people that are exposed to (and/or ultimately at risk of) extreme flooding. They do so by investigating limitations in accuracy and resolution in existing population distribution maps. Their efforts need to be applauded, and the results are quite stunning regarding the amount of uncertainties they found, which will likely have major ramifications for existing and future studies relying on these kinds of data. This issue has been speculated about in the past, but, to my knowledge, this study provides the most convincing quantification of the problem at a global scale. The findings would represent a major improvement in our attempts to produce reliable information for flood mitigation and management strategies.

That being said, I am afraid I cannot support publication of the manuscript in its current form, in particular in such a high-level journal as Nature Communications, for one major reason: The entire study hinges on the assumption that the new High Resolution Settlement Layer (HRSL) is superior in quality to any other global population dataset. However, there is very little (if any) evidence provided in the manuscript to support this assumption. The text only states that “The final [HRSL] density maps were validated against ground-truth household surveys taken from the World Bank Living Standards Measurement Study (LSMS)” (lines 265-267), but no specifics or quantitative results from this comparison are mentioned. I tried looking up this dataset and could not find any relevant scientific publication regarding its validity. All I could find was a short project description on a Facebook blogpost from 2016 (<https://code.facebook.com/posts/596471193873876>), stating that: “Our collaborators at the World Bank validated the Malawi dataset and found that on average we miss only about 6% of the houses that are part of their 2011 household survey data.” This was followed by a broken link and I could not find any additional information. I am afraid that the lack of presented validation of HRSL (either by using citations or by own efforts by the authors) renders the study not publishable in its current form. Please do not misunderstand: I am NOT assuming that HRSL is wrong; I actually believe it is very likely the better dataset. But this needs to be shown with proper scientific evidence rather than just be assumed.

Here is as an example of why I think a solid verification of HRSL is indispensable: As Smith et al. show, a main feature of HRSL seems to be that it registers far lower population densities in rural areas, i.e. mostly zero, while other datasets show mostly small but non-zero values in rural areas. It might be possible that this is, at least in part, caused by the detection methods: HRSL is based on remote sensing imagery from which individual buildings are detected, and this may lead to higher

uncertainties in very low density areas. With zero values in rural areas in HRSL, and assuming that the total population numbers for districts or countries are kept the same as in other datasets, the total population would shift towards urban areas by default. This shift of rural to urban population (true or not) would explain some of the results presented in the study by Smith et al. Hence testing whether the zero rural population numbers are indeed correct in HRSL is a fundamental issue for the validity of the presented study.

Many thanks for the above observations. As also highlighted by the editor, a validation of the HRSL dataset is indeed indispensable. The Facebook team have recently published such a validation via [arXiv \(https://arxiv.org/\)](https://arxiv.org/), before full publication in a peer reviewed journal article later this year. The paper outlines several metrics used to validate the HRSL dataset and should address the reviewers concerns with regards to validation:

Tiecke, T.G., Liu, X., Zhang, A., Gros, A., Li, N., Yetman, G., Kilic, T., Murray, S., Blankespoor, B., Prydz, E.B., Dang, H.-A.H., 2017. Mapping the world population one building at a time. CoRR [abs/1712.05839](https://arxiv.org/abs/1712.05839).

The above paper outlines numerous validation exercises, including the testing of building identification using the Malawi Third Integrated Household Survey (IHS3). This survey recorded the location of >11,000 households nationwide and is thus independent of remote sensing methods. When used as a validation dataset, the results revealed that 98.3% of IHS3 household locations coincided with HRSL populated cells/pixels.

A separate analysis of 3 different population datasets was undertaken for a single region near Blantyre, Malawi, where buildings were manually identified and labelled. A comparison of HRSL, GUF and GHSL revealed that in urban areas 99%, 82% 83 % of buildings were identified correctly. However, in rural areas the recall values were 82%, 4% and 6% for HRSL, GHSL and GUF datasets respectively. The results indicate that HRSL has a far superior performance in rural areas, suggesting that the non-zero population numbers in rural areas are indeed robust.

There are a two other larger issues I would like to mention, yet they seem easier to be addressed:

a) Lines 144-145: "HRSL demographic data distributes populations across a far smaller area, resulting in much higher population densities". And lines 221-224: "The HRSL returns larger concentrations of risk, suggesting that although there is a reduction in the total number of people exposed, these populations are confined to a smaller area, constituting higher concentrations of exposure." This is a very interesting characteristic of the HRSL data and indicates that likely HRSL will show a strong sensitivity when intersected with flood risk pixels; i.e. a single pixel of additional inundation extent can trigger an enormous addition of population exposed (which is not the case if population data is more smoothly distributed). This raises the question whether the application of HRSL data can lead to increased total uncertainties when intersected with the inherent spatial uncertainty of inundation data. It would be very interesting to conduct some kind of sensitivity analysis by applying uncertainty boundaries in the inundation data and see what the effect is in the total exposure results. This is particularly relevant as the manuscript states that the employed flood

modelling framework “captured between two thirds and three quarters of the area determined to be at risk” (lines 252-254). What if the missing third or quarter of flood areas has high population singularities in them? Would this increase the numbers of total exposed population to be similar to those of traditional population datasets?

Again, many thanks to the reviewer for their comments. We agree entirely that some form of sensitivity analysis would be a very beneficial to the papers findings. We have therefore included a sensitivity analysis changing both the resolution of the population and hazard data, and a separate sensitivity analysis exploring changing recurrence interval. The results of both additional pieces of analyses revealed that firstly, reductions in the amount of people exposed to flooding are only seen when you have both high-resolution population data and hazard data. Indeed, this additional analysis allowed us to draw conclusions about the significance of high resolution hazard data; the results revealed that exposure estimates escalate rapidly with coarser resolution hazard data.

The exposure estimates calculated across different return periods revealed that the differences in exposure estimates between each population dataset remain largely consistent. HRSL always returns the lowest estimate of population exposed. Moreover, the differences between the exposure estimates deviate further with increasing return period. We therefore do not see large increases in exposure as we change the magnitude of flooding, as hypothesised. The results suggest that exposure estimates conducted using HRSL aren't particularly sensitive to changing recurrence interval and the conclusions drawn in the paper appear to be robust to the inherent uncertainties in the hazard data.

b) In the methods section, it is not explained how the intersection between the 30m pixels of the flood model and the coarser 90m and 900m pixels of the WorldPop and LandScan population datasets was conducted. Line 284 only indicates that population values of wet pixels were summed. Were the coarser pixels first disaggregated to the finer 30m resolution? If no, this would be a major problem. If yes, how was the disaggregation conducted? In particular: was it assumed that population density is homogeneous within each coarser pixel? And if yes, what implications would such an assumption have? This could be tested by re-calculating the results, yet assuming that population numbers within the coarser resolution pixels are NOT equally distributed within the pixel, but people are located exclusively in the non-flood parts of the pixel. This could easily be achieved by simply removing all original WorldPop and LandScan pixels from the analysis that are only partially flooded. If this would lead to results similar to those delivered by HRSL, then the problem is not the accuracy of WorldPop and LandScan data, but only their coarser resolution (whereas currently the authors seem to indicated that the problem lies in both accuracy and resolution).

As the reviewer indicates, we homogenously distributed the coarser resolution population data amongst the higher resolution cells. The implications of this have been explored in the resolution sensitivity analysis, described previously. The results show that even when the HRSL is aggregated up to ~90m and ~900m, it always returns the lowest exposure estimates across the population datasets used. These results indicate that not only the resolution, but also the accuracy of the HRSL dataset that is responsible for differences seen in total exposure estimates.

And here are some additional minor/detailed comments:

- The text is generally well written and clear, but could be a little more concise in parts. For any

revision or new submission elsewhere, I encourage the authors to carefully check the manuscript and remove any redundant or repetitive statements, sentences, or parts of sentences.

Hopefully the revised text is more concise.

- The definitions of exposure, vulnerability, risk, and hazard are not entirely clear in the manuscript. It seems that sometimes “exposure data” is used synonymous with “population data”. A bit more explanation would be helpful.

Agreed. The manuscript has been corrected, using hazard and exposure to describe the flood and population data respectively. The intersection of flood data and exposure data has been termed risk. The definition of ‘risk’, which in this case omits any vulnerability component, has been outlined in the methods section:

“In this study, risk is defined by the intersection of hazard (flood model output) and exposure (population) data, the way in which hazard and exposure data interact (vulnerability) has not been considered”

- Line 24: you say that pixels at a resolution of 1-30 arc seconds have an area of ~30-900 square meters at the equator. You need to square the numbers if you provide results in square meters, i.e. 1-30 arc seconds equals 900-810,000 square meters.

We’ve changed the description to ‘horizontal resolution’.

- Line 84: Replace “of volume” with “volume”

Corrected.

- Line 120: The HRSL data is suddenly called “Facebook data”. Be consistent.

Corrected.

- Line 122 (and possibly elsewhere): the use of the word “significantly” implies some statistical metric has been used (i.e. statistically significant), but I guess this is not the case here; use other terminology to avoid confusion, or provide the statistical metrics.

Changed to ‘markedly’.

- In Table 2, the HRSL column is presented twice. This could be avoided if HRSL becomes the first column, and then WordPop and LandScan numbers are provided, each with absolute numbers followed by percent difference.

Table has been edited.

- Figure 2: Legend of panels A-C is confusing. It seems on panel A there are a lot of NoData pixels. How is NoData different from the lowest color (yellow) which seems to indicate zero population? I

guess the legend statement “0” does not really mean zero but some range that is larger than zero? And the other colors represent ranges as well?

Yes the 0 in the legend does not mean zero; in the case of the WorldPop and LandScan data, this is in fact a very small population density value.

- Line 157 (and elsewhere): I am not sure whether the distinction of urban vs. rural areas should be simply labeled as “land use types” (I typically expect many more classes in land use classifications, e.g. agricultural techniques, recreational use, etc.). Is there a better expression?

Good question. Unfortunately, the Global Human Settlement (GHS) Layer only partitions land use into 3 categories at the global scale. These are the three categories that have been included here. In reality, of course there will be more complexity in the differentiation between land use classifications, unfortunately these are not represented in the GHS.

- Lines 168-174 (and Figure 4): I guess what you describe here as “exposed” areas are areas where inundation AND non-zero people intersect in a pixel, right? This is not entirely clear as the land area would still be exposed even if there are no people living there.

We agree that this was somewhat confusing. We have elaborated further:

Figure 4 reveals significant differences in the areas deriving flood risk, with >95% of the total exposed area (area where both hazard and population data are non-zero)

- Lines 190-192: You say that “Critically, it is exposure hotspots in new urban centres that appear to be driving increased flood losses in developing regions.” I cannot see any evidence in your study that supports this statement. Is this a literature citation? If not, you need to explain where this statement comes from.

This is a reference to Di Baldassarre *et al.* (2010), not the work conducted in this study. The manuscript has been edited to reflect this.

- Line 217: Replace “spatially” with “spatial”

Corrected

- Line 215: Remove “t”

Corrected

- Line 265: Replace “produced” with “produce”

Corrected

Reviewer 3

Review of "New estimates of flood exposure in developing countries using high-resolution population data" by Smith et al.

I really enjoyed reading this paper and find that it has valuable contributions to our understanding of flood exposure estimates. It provides detailed information on the total exposure of people in of 15 developing countries at an unprecedented spatial resolution. Clearly the HRSL population data provides additional information that is needed to compute the flood exposure. Overall, this is a nice manuscript, but I have one major point of concern and that is towards the degree of novelty of this work.

In the beginning of the manuscript the authors give the impression that they developed a 90m population dataset and then used their flood model to give flood exposure estimates. After careful reading I find that the map is created by Facebook, Columbia and the World Bank. This clearly takes away most of the novelty of this work, since now the authors use their existing high-resolution flood model and overlay it with an new population map. Although I can see that this provides a novel dataset that has not been created before (as correctly addressed by the authors in Table 1) it is not extreme (technically) difficult and/or novel. Previous studies did the same experiment but then at coarser spatial resolution. The HRSL population data is clearly novel and so is the exposure map, but it can be seen as incremental science compared to earlier flood exposure estimates by the same group (<http://www.fathom.global/flood-maps>, scroll down toward the bottom). They already have their 90m

flood model (with flood defenses as I can find on the website, not in the paper, maybe add this information or I missed it) and then it is a simple overlay operation. Maybe I'm completely wrong, in that case I think the authors would find it easy to convince me and potential reader.

Although I state above that I have some problem with the degree of scientific novelty, I can clearly see an add value for the world community. If the authors agree with me that this is the major selling point of the manuscript, I feel that this analysis can easily be done at the global scale. The authors already have the global flood maps at 90m, they have the HRSL population data at global scale, so I see no good reason why they shouldn't do that. This would also give nice figures (fancy Nature world map figures), and can also inform the audience on where we have currently under/overestimated the flood exposure. The authors indicate that the largest difference can be found in the rural areas, so I would expect substantial reductions in flood exposure in, for example, China.

To summarize, it is a sound manuscript, clearly written and a good story. I can see the added value of this manuscript, but I would just hope that it would provide either some more novel computation or analysis to make it a Nature Communications paper. I have provided some suggestions below:

- Do a global analysis

Unfortunately, the HRSL dataset has only been produced for a limited number of countries, precluding a global scale analysis, which of course would be a valuable future contribution.

- Perform the same analysis (if possible with the flood model input), for different return periods

This is a very useful suggestion and has been included in the revised manuscript.

- Perform the same analysis with and without flood defences

This would be a very nice addition, unfortunately it is precluded by the availability of robust global scale flood defence data.

- Perform the same analysis but then for two different time periods. One historical vs current population, showing an increase or decrease in exposure.

This is unfortunately precluded by the availability of each of the population datasets at different time periods. However, some population datasets do exist for historical periods and is something we feel would have enough content to make for a separate publication.

Minor comments:

- It should be stated more clearly that the HRSL dataset is an existing dataset, the manuscript now give the impression that it is created in this manuscript.

Hopefully this is clearer in the text.

- Line 120, there is a Facebook reference, which should be HRSL

Corrected.

- Line 166-167, I'm not entirely sure if we should call this a negative correlation, since it only has three points on which this statement is based (the three different datasets). Maybe rephrase to negative relationship

Corrected.

- Did the author validate the population or flood model? Is there a way to verify if the population map is accurate and or if the flood maps are accurate? I can see that it can be difficult to validate the flood maps, but are there some partial validations available?

The flood hazard model has been validated by Sampson et al., (2015). This validation exercise has been included in the Methods section.

A validation of the HRSL dataset has been conducted by Tiecke et al., (2017). This has been included in the revised manuscript.

- Figure 1, could be improved by using a log scale or something, now it is difficult to see the blue line and the divergence between the lines for the high exposure values

Agreed. We have revised the figure, which is hopefully clearer.

- Maybe the author can make a world map figure with a pie chart per country? Then the reader has a visual on where the biggest changes occur

Niko Wanders, Utrecht University

Reviewers' comments:

Reviewer #2 (Remarks to the Author):

Dear Editor and Authors,

I have been Reviewer #2 on the original manuscript by Smith and colleagues: "New estimates of flood exposure in developing countries using high-resolution population data". As such, my original comments about the importance of this study remain the same: I still believe that the presented findings represent a major improvement in our attempts to better quantify flood risks at large scales (by refining estimates of exposed population) and to ultimately produce more reliable information for flood mitigation and management strategies.

After reading the revised manuscript I am very pleased that the authors added several major new contributions, including more discussion regarding the quality of their population data; different return periods for the flood inundation data; as well as a basic sensitivity analysis testing the effects of spatial resolution. All of this clearly adds robustness and rigor to their study, which is much appreciated.

In their rebuttal, the authors sufficiently addressed most of my original comments; although I have a few follow-up comments below. There is one major issue, however, for which I look towards the Editor for a decision: in my original review my main concern related to the lack of evidence regarding the quality and evaluation of the new population data. As reviewer #3 pointed out as well: the real novelty of the presented study lies in the application of this new population data; and thus inherently depends on its quality. The authors added a more thorough discussion of the new data in the Methods section, and they also list one reference for it: Tiecke et al. (2017). This reference is an unpublished manuscript available at arXiv. While I appreciate this document, it also confirms that the data did not yet undergo a thorough peer review process. I am familiar with the common approach of researchers to use new data as soon and quickly as possible (I do so myself) even if the corresponding publication is still in progress. But typically this is done only if the data producers are part of the author team, and/or for less critical applications of the data in question. In the case of this manuscript, the analysis depends entirely on the new population dataset that is created by an independent group of researchers. As a reviewer I thus feel that I need to voice my concern about this, yet I will leave it to Nature Communications as to whether they can agree to publish an application of data which is not yet officially peer reviewed. Maybe there is a chance to wait for the population data to be published first?

Aside from this concern, I think the manuscript is generally well written and structured. Yet I had the feeling that in particular new and added parts were sometimes less polished in their wording than the original text; hence my new list of detailed comments below is actually longer than in the original revision. There are quite a few minor typos, e.g. inconsistent use of spaces between numbers and units ("90m" vs. "90 m"), inconsistent spelling, etc. I encourage the authors to give this manuscript another good read before publication.

Detailed comments:

- Abstract and elsewhere in text: The spelling of "data set" vs. "dataset" is inconsistent.
- Line 13: I guess this should read "single hazard data set" rather than "single exposure data set". Otherwise the next sentence makes no sense which claims that the effect of exposure data sets has not been analyzed before.
- Line 59: Replace "is ~3..." with "is the ~3..."
- Lines 112-113: This sentence should state somewhere that the bias is in comparison to HRSL.

- Lines 116-118: This entire sentence seems to be repetitive of the previous sentence, just using LandScan instead of WorldPop. I suggest combining the two sentences to avoid redundancy.
- Line 153: Remove apostrophe after floodplain.
- Line 165: It would help to call the events "flooding events".
- Line 166: I had already raised concerns in my original review about using the expression "land use" to refer to the level of urbanization. I think it would be easy to avoid this confusion. For example, the expression "across different land use types revealed..." could be replaced with "was differentiated into urban, semi-urban, and rural areas and revealed...".
- Lines 166-169: There is currently no information provided as to how the authors have differentiated the urbanized vs. rural zones. I see that this is explained in the Methods section, but it would help to at least hint at this in the main text, and/or add a citation, or at least point to the Methods section for more detail.
- Line 167-168: Replace "... exposed; exposed rural populations make up 47%" with "... exposed representing 47%".
- Line 177: Replace "areas deriving flood risk" with "areas exposed to flood risk".
- Lines 198ff: The entire Discussion section contains quite a lot of new arguments. It seems that the style of this section is somewhat less refined than other sections, with increasing typos and some ambiguous expressions and explanations. I make some comments below, but the authors should pay particular attention to this section before publication.
- Line 202: Avoid colloquial phrase "aren't".
- Line 203: Remove comma.
- Line 204: I do not really understand what this means: "are expected to be robust to uncertainties".
- Line 206: Avoid colloquial expression "only when you have" (the entire sentence could be reworded).
- Lines 209-210: If the increase in population is quantified (doubling) then the corresponding coarsening needs to be quantified as well (i.e., which resolution change causes this doubling); otherwise the statement is not really meaningful.
- Lines 210-212: "Even when the HRSL data is aggregated to the resolutions of WorldPop and LandScan™ data, we see the same trends as those detected when calculating risk at their native resolutions." – I would expect a more cautious wording here. The trends are similar only in that HRSL numbers are still smaller. But the actual differences got a lot smaller; i.e. they are not "the same". This result indicates that resolution plays at least a partial role.
- Lines 286-287: Here and elsewhere, the authors try to define their use of the expressions exposure, hazard, risk, and vulnerability. In particular, they say: "In this study, risk is defined by the intersection of hazard (flood model output) and exposure (population) data". Also, vulnerability is defined as "the way in which hazard and exposure data interact". Unfortunately, I find these definitions to be at odds with most of the literature that I know of. According to the UN Office for Disaster Risk Reduction (UNISDR 2009), "exposure is defined as the people, property, systems, or other elements present in hazard zones that are thereby subject to potential losses". This equates

to what the authors define to be 'risk' in their definition above. Also, the authors still seem to equate any population map to represent 'exposure' (see here and also lines 99-100), independent of whether these populations are in the flood zone or not. To assess risk, exposure (population in floodplains) needs to be multiplied (not simply intersected) with hazard (e.g. multiply by flooding depth; i.e. a 5 meter flood creates more risk than a 1 meter flood) and also with vulnerability (which the authors decided to neglect in their definition, ok). To me, it remains very confusing to equate 'risk' with 'exposed population', as the authors do in Table 2. Note that I have no issue to call the exposed population within the flooding zones 'exposed population' or 'population at risk', because 'at risk' is different from the quantified term 'risk' itself.

- Line 305 (and earlier): The acronyms GHSL (Global Human Settlement) and GUF (Global Urban Footprint) are introduced and referenced in line 28. In line 104, a "Global Human Settlement Layer" is explained but not referenced, i.e. it is not clear whether this is GHSL. Finally, the acronyms GHSL and GUF reappear in the Methods section in line 305, without further explanation of the data or any reference (e.g., there is no information on their resolution or their source of information). I would have expected more explanation about this in the Methods section.

- Lines 305 and 307: Why is the sequence between GHSL and GUF flipped?

- Line 306: What are "recall values"?

- Line 310: "population" instead of "populations".

- Line 311: Spelling of "night-time" is with hyphen here; but it was without hyphen in line 30.

- Line 313: The expression "precise" is not a good fit here (it is not referring to data precision).

- Lines 325-326: "... by homogenously distributing the population totals at a coarser resolution, amongst the higher resolution cells." This wording is not very clear and should be improved.

- Line 328: Was the scaling performed for country values? Or for global totals? Or for other units?

- Line 378, Reference 17: This is not a properly formatted citation.

- Table 2: In the column headers, "change" should be replaced with "difference".

- Figure 2: In my original review, I had already commented that I find the legend of panels A-C confusing. In their rebuttal letter, the authors clarified what the legend is supposed to show, but they did not make any changes in the actual paper. The figure legend still shows values of "0" (in yellow) which in reality means "non-zero". Also, the legend shows only a few explicit class values instead of class ranges, so it remains unclear in which class a value of, say, 10 would fall (color of 0 or 25?). This could all be easily resolved by updating the labels in the legends to make the colors unambiguous. The yellow color should have a precise range, e.g. ">0 – 25". In addition, it would be very helpful to break out one additional color for very small values, e.g. ">0 – 1", as the lowest class is very important but seems to cover a large range of values (>0 – 25?).

Reviewer #4 (Remarks to the Author):

The authors present an analysis of intersection of a newly published population dataset (HRSL) with global flood hazard maps. The results show that the risk estimated using HRSL data is significantly lower than what has been estimated earlier with other datasets including the LandScan and WorldPop. The authors also performed sensitivity analysis using multiple resolutions of both flood hazard data and population data to solidify their findings. Overall, I think this is a

very useful study, but I am not sure about the novelty of scientific contribution. Basically the authors have taken two existing datasets (HRSL) and flood hazard layer and found the number of houses within their intersection. As the authors point out, this has already been done earlier using coarser resolution LandScan and WorldPop data. The results basically say that using a higher resolution population data gives you lower risk compared to coarser resolution data. Additionally, as you decrease the resolution of both HRSL and flood hazard data, the discrepancy in the results reduces.

I am also not convinced that just intersecting the population layer with flood hazard layer will give you risk. The analysis lacks the information on any flood protection in place that will play significant role in defining the risk. If the results show high risk in urban areas, it is likely that many of these areas may have some flood defense and so the number of people affected may be lower. Alternatively, the smaller number of rural communities exposed to flood hazard may be at higher risk in the absence of any flood protection.

I would have considered this study more novel if the authors had gone beyond the intersection of two spatial datasets and incorporated additional information or analysis that the previous analysis missed to provide a more robust risk analysis. For this reason, I do not recommend the publication of this article in Nature Communications.

Reviewer #2

We would like to thank the reviewer for their contribution, they have clearly put considerable effort into reviewing our work and their input has undoubtedly resulted in a stronger paper.

The main point raised related to the validation of the HRSL dataset. On the recommendation of the Editor, we have undertaken our own review of the HRSL dataset, which has been included in the Supplementary Material.

I have been Reviewer #2 on the original manuscript by Smith and colleagues: “New estimates of flood exposure in developing countries using high-resolution population data”. As such, my original comments about the importance of this study remain the same: I still believe that the presented findings represent a major improvement in our attempts to better quantify flood risks at large scales (by refining estimates of exposed population) and to ultimately produce more reliable information for flood mitigation and management strategies.

After reading the revised manuscript I am very pleased that the authors added several major new contributions, including more discussion regarding the quality of their population data; different return periods for the flood inundation data; as well as a basic sensitivity analysis testing the effects of spatial resolution. All of this clearly adds robustness and rigor to their study, which is much appreciated.

In their rebuttal, the authors sufficiently addressed most of my original comments; although I have a few follow-up comments below. There is one major issue, however, for which I look towards the Editor for a decision: in my original review my main concern related to the lack of evidence regarding the quality and evaluation of the new population data. As reviewer #3 pointed out as well: the real novelty of the presented study lies in the application of this new population data; and thus inherently depends on its quality. The authors added a more thorough discussion of the new data in the Methods section, and they also list one reference for it: Tiecke et al. (2017). This reference is an unpublished manuscript available at arXiv. While I appreciate this document, it also confirms that the data did not yet undergo a thorough peer review process. I am familiar with the common approach of researchers to use new data as soon and quickly as possible (I do so myself) even if the corresponding publication is still in progress. But typically this is done only if the data producers are part of the author team, and/or for less critical applications of the data in question. In the case of this manuscript, the analysis depends entirely on the new population dataset that is created by an independent group of researchers. As a reviewer I thus feel that I need to voice my concern about this, yet I will leave it to Nature Communications as to whether they can agree to publish an application of data which is not yet officially peer reviewed. Maybe there is a chance to wait for the population data to be published first?

As indicated above, we have included a validation of the HRSL data undertaken using Open Street Map data. This validation work will be included in the Supplementary Material.

Aside from this concern, I think the manuscript is generally well written and structured. Yet I had the feeling that in particular new and added parts were sometimes less polished in their wording than the original text; hence my new list of detailed comments below is actually longer than in the original revision. There are quite a few minor typos, e.g. inconsistent use of spaces between numbers and units (“90m” vs. “90 m”), inconsistent spelling, etc. I encourage the authors to give this manuscript another good read before publication.

We apologise for the standards. We have read the manuscript through and have corrected all the typos that have been indicated. Many thanks to the reviewer for pointing these out.

- Abstract and elsewhere in text: The spelling of “data set” vs. “dataset” is inconsistent.

Corrected.

- Line 13: I guess this should read “single hazard data set” rather than “single exposure data set”. Otherwise the next sentence makes no sense which claims that the effect of exposure data sets has not been analyzed before.

The wording is correct; the point is a reference to Trigg et al. in which a single exposure dataset was used to explore the number of people living in hazardous areas, using an ensemble of large-scale flood hazard models. The paper subsequently found wide variation between each of the models, even when the same exposure dataset was used.

- Line 59: Replace “is ~3...” with “is the ~3...”

Corrected.

- Lines 112-113: This sentence should state somewhere that the bias is in comparison to HRSL.

Corrected.

- Lines 116-118: This entire sentence seems to be repetitive of the previous sentence, just using LandScan instead of WorldPop. I suggest combining the two sentences to avoid redundancy.

Corrected.

- Line 153: Remove apostrophe after floodplain.

Corrected.

- Line 165: It would help to call the events “flooding events”.

Corrected.

- Line 166: I had already raised concerns in my original review about using the expression “land use” to refer to the level of urbanization. I think it would be easy to avoid this confusion. For example, the expression “across different land use types revealed...” could be replaced with “was differentiated into urban, semi-urban, and rural areas and revealed...”.

Corrected.

- Lines 166-169: There is currently no information provided as to how the authors have differentiated the urbanized vs. rural zones. I see that this is explained in the Methods section, but it would help to at least hint at this in the main text, and/or add a citation, or at least point to the Methods section for more detail.

We've added a link to the methods section as well as a little more detail on the analysis in the Methods section.

- Line 167-168: Replace "... exposed; exposed rural populations make up 47%" with "... exposed representing 47%".

Corrected.

- Line 177: Replace "areas deriving flood risk" with "areas exposed to flood risk".

Corrected.

- Lines 198ff: The entire Discussion section contains quite a lot of new arguments. It seems that the style of this section is somewhat less refined than other sections, with increasing typos and some ambiguous expressions and explanations. I make some comments below, but the authors should pay particular attention to this section before publication.

- Line 202: Avoid colloquial phrase "aren't".

Corrected.

- Line 203: Remove comma.

Corrected.

- Line 204: I do not really understand what this means: "are expected to be robust to uncertainties".

The results of the analysis and the conclusions drawn have remained consistent across different recurrence intervals. As the results remain the same across different return periods, this suggests that the conclusions drawn here are robust to uncertainties in the hazard model, as the different recurrence interval footprints essentially represent different realisations of flood hazard. We've added a little extra text to hopefully make this clearer.

- Line 206: Avoid colloquial expression "only when you have" (the entire sentence could be reworded).

Corrected.

- Lines 209-210: If the increase in population is quantified (doubling) then the corresponding coarsening needs to be quantified as well (i.e., which resolution change causes this doubling); otherwise the statement is not really meaningful.

Agreed, corrected.

- Lines 210-212: "Even when the HRSL data is aggregated to the resolutions of WorldPop and LandScan™ data, we see the same trends as those detected when calculating risk at their native resolutions." – I would expect a more cautious wording here. The trends are similar only in that HRSL numbers are still smaller. But the actual differences got a lot

smaller; i.e. they are not “the same”. This result indicates that resolution plays at least a partial role.

Agreed, corrected.

- Lines 286-287: Here and elsewhere, the authors try to define their use of the expressions exposure, hazard, risk, and vulnerability. In particular, they say: “In this study, risk is defined by the intersection of hazard (flood model output) and exposure (population) data”. Also, vulnerability is defined as “the way in which hazard and exposure data interact”. Unfortunately, I find these definitions to be at odds with most of the literature that I know of. According to the UN Office for Disaster Risk Reduction (UNISDR 2009), “exposure is defined as the people, property, systems, or other elements present in hazard zones that are thereby subject to potential losses”. This equates to what the authors define to be ‘risk’ in their definition above. Also, the authors still seem to equate any population map to represent ‘exposure’ (see here and also lines 99-100), independent of whether these populations are in the flood zone or not. To assess risk, exposure (population in floodplains) needs to be multiplied (not simply intersected) with hazard (e.g. multiply by flooding depth; i.e. a 5 meter flood creates more risk than a 1 meter flood) and also with vulnerability (which the authors decided to neglect in their definition, ok). To me, it remains very confusing to equate ‘risk’ with ‘exposed population’, as the authors do in Table 2. Note that I have no issue to call the exposed population within the flooding zones ‘exposed population’ or ‘population at risk’, because ‘at risk’ is different from the quantified term ‘risk’ itself.

We agree with the reviewer’s comments here. The authors did debate upon which terminology to use throughout the paper, as there is some variation in the literature. However, on reflection we agree that any indication that we have undertaken any ‘risk’ calculations is mis-leading. We have altered the text to define all our calculations as calculations of *exposure* and not *risk*. We have also removed any reference to the population datasets being exposure datasets.

- Line 305 (and earlier): The acronyms GHSL (Global Human Settlement) and GUF (Global Urban Footprint) are introduced and referenced in line 28. In line 104, a “Global Human Settlement Layer” is explained but not referenced, i.e. it is not clear whether this is GHSL. Finally, the acronyms GHSL and GUF reappear in the Methods section in line 305, without further explanation of the data or any reference (e.g., there is no information on their resolution or their source of information). I would have expected more explanation about this in the Methods section.

Corrected

- Lines 305 and 307: Why is the sequence between GHSL and GUF flipped?

Corrected

- Line 306: What are “recall values”?

Corrected

- Line 310: “population” instead of “populations”.

Corrected

- Line 311: Spelling of “night-time” is with hyphen here; but it was without hyphen in line 30.

Corrected

- Line 313: The expression “precise” is not a good fit here (it is not referring to data precision).

Corrected

- Lines 325-326: “... by homogenously distributing the population totals at a coarser resolution, amongst the higher resolution cells.” This wording is not very clear and should be improved.

Corrected

- Line 328: Was the scaling performed for country values? Or for global totals? Or for other units?

Scaling was undertaken at the country level. Included in the text.

- Line 378, Reference 17: This is not a properly formatted citation.

Corrected

Reviewer #4

The authors present an analysis of intersection of a newly published population dataset (HRSL) with global flood hazard maps. The results show that the risk estimated using HRSL data is significantly lower than what has been estimated earlier with other datasets including the LandScan and WorldPop. The authors also performed sensitivity analysis using multiple resolutions of both flood hazard data and population data to solidify their findings. Overall, I think this is a very useful study, but I am not sure about the novelty of scientific contribution. Basically the authors have taken two existing datasets (HRSL) and flood hazard layer and found the number of houses within their intersection. As the authors point out, this has already been done earlier using coarser resolution LandScan and WorldPop data.

Many thanks for these comments. We're glad that Reviewer #4 feels that the study is a useful one., but we would defend, quite strongly in fact, the novelty of the scientific contribution. The fact that coarse population data sets have previously been intersected with flood hazard layers doesn't mean that a similar analysis cannot lead to new and original conclusions. Indeed, an exposure analysis is a fundamental calculation in natural hazards science and not something over which past papers have a monopoly. Rather it is the conclusions that result from the analysis that give rise to the novel contribution. In this case, there are good reasons to suspect that the results from previous exposure analyses will contain significant biases. Testing this hypothesis can only be accomplished by performing a new exposure calculation with better data, and this is exactly what this paper does. The new result is to quantify the likely scale of these biases, and its significance stems from the fact that such biases will occur in all large-scale flood risk analyses published to date. Further, such biases will likely be present in the risk analyses for all other natural hazards too.

The results basically say that using a higher resolution population data gives you lower risk compared to coarser resolution data. Additionally, as you decrease the resolution of both HRSL and flood hazard data, the discrepancy in the results reduces.

This is correct, and these conclusions, which will be important for a large community of researchers and decision makers, can only be arrived at with the data sets and through the analysis reported in this paper. Our view is that these comments clearly indicate just why this paper should be published.

I am also not convinced that just intersecting the population layer with flood hazard layer will give you risk.

This is a good point, and we perhaps should have been more precise in our terminology. To be clear, intersecting the population layer with the flood hazard layers gives the exposure, or more specifically in this case the 'population at risk'. We have changed 'risk' to 'exposure' at appropriate points throughout the paper.

The analysis lacks the information on any flood protection in place that will play significant role in defining the risk. If the results show high risk in urban areas, it is likely that many of these areas may have some flood defense and so the number of people affected may be lower. Alternatively, the smaller number of rural communities exposed to flood hazard may be at higher risk in the absence of any flood protection.

Actually, flood defences are taken into account in our flood inundation modelling. We develop multivariate regression relationships between flood defence standards of protection for known locations and national scale socio-economic data. This allows us to predict likely flood protection standards for locations where this is not known. These estimates are used to increase river channel capacity for different scales of urban areas, with the location of urban areas identified using satellite night time light data. In practice, the developing countries considered in this study largely lack significant flood protection infrastructure, even for large settlements. The impact of any uncertainties in our method are therefore likely to have only a negligible impact on the results obtained and the conclusions drawn.

I would have considered this study more novel if the authors had gone beyond the intersection of two spatial datasets and incorporated additional information or analysis that the previous analysis missed to provide a more robust risk analysis. For this reason, I do not recommend the publication of this article in Nature Communications.

We would argue, again quite strongly, that the sensitivity and scale analysis we have performed is a substantive technical contribution and does provide a robust analysis. Whilst intersecting the data layers and performing the scale and statistical analysis is written as if relatively straightforward in order to make the paper accessible to a wide readership, this is in fact not the case. Instead, performing the country-scale geographical analysis is a substantial coding and computational undertaking. Moreover, the precursor technical work to develop a global hydrodynamic model with proven local skill is huge and perhaps under-appreciated by the reviewer.

REVIEWERS' COMMENTS:

Reviewer #2 (Remarks to the Author):

Dear Editor and Authors,

I have been Reviewer #2 on the original and revised manuscripts by Smith and colleagues: "New estimates of flood exposure in developing countries using high-resolution population data".

I think the authors have sufficiently addressed my comments and concerns that I raised in my last revision, and I am happy to suggest publication of their manuscript. I particularly like that they changed the terminology from "risk" to "exposure", and that they added their own validation of the HRSL data.

I still saw a few typos and occasional minor formatting issues (of which I list some random ones below), but I assume that the editor and/or authors will carefully go over the proofs anyway.

- Lines 61-62: you end your listing of different population datasets with the highest resolution one (only available for the US), which is at odds with the next sentence that starts by referring to "this coarse resolution".
- In Line 99, you introduce the acronym HRSL, i.e. "called the High Resolution Settlement Layer (HRSL)", yet you already used the acronym earlier.

In the Supplementary Material:

- "74,000 buildings" instead of "74000 building"
- Should table captions not be above the tables?
- Table 3 is not referenced in the text.
- The equations should be numbered (Eq. 1, etc.)
- I find the tone still a little colloquial in places (e.g. "struggle"; "can never achieve this"; also, can an algorithm make "sensible choices"?).
- And finally a subjective comment about the Supplementary Material: I sense that there might be a bit of a dismissive tone against the GPW and LandScan datasets, which are described to have "little to no skill" and "can never achieve" the quality of HRSL. I think the data limitations are obvious, and the datasets are not claiming that they are suitable at those high resolutions. So the focus should really stay on HRSL here, as the main question is: how good is HRSL (not how bad are GPW and LandScan). In any case, I think the validation results regarding HRSL are convincing enough to go ahead with the study.

Reviewer #4 (Remarks to the Author):

My primary concern with this contribution was its novelty. I think the authors have given a convincing answer for why this contribution is novel. I am okay with publishing this article in nature communications.